REGISTERED REPORT PROTOCOL

# A registered report of a crossover study on the effects of face masks on walking adaptability in people with Parkinson's disease and multiple sclerosis

**Mareike Eschweiler[1,2], Christopher McCrum[3,4], Eleftheria Giannouli[5,6]** *

1 Neurological Rehabilitation Center Godeshoehe GmbH, Department of Therapeutic Science, Bonn, Germany, 2 Department of Epidemiology, CAPHRI Care and Public Health Research Institute, Maastricht University, Maastricht, The Netherlands, 3 Department of Nutrition and Movement Sciences, NUTRIM School of Nutrition and Translational Research in Metabolism, Maastricht University, Maastricht, The Netherlands, 4 Neuromotor Rehabilitation Research Group, Department of Rehabilitation Sciences, KU Leuven, Leuven, Belgium, 5 Department of Health Sciences & Technology, Institute of Human Movement Sciences and Sport, ETH Zurich, Zurich, Switzerland, 6 Division of Sports and Exercise Medicine, Department of Sport, Exercise, and Health, University of Basel, Basel, Switzerland

☯ These authors contributed equally to this work.
* eleftheria.giannouli@hest.ethz.ch

## Abstract

### Background

Face masks protrude into the lower visual field causing reduced perception of visual stimuli, potentially making obstacle avoidance during walking more difficult and increasing fall risk. Recommendations on walking and mask wearing for older adults have been debated, with no clear consensus on the various factors interacting and influencing walking safety while wearing a face mask. It is particularly important to address this issue in populations at an increased risk of falls. Therefore, this study aims to investigate the effects of mask-wearing on objectively measured walking adaptability in people with Parkinson's disease and Multiple Sclerosis.

### Methods

50 patients with either Parkinson's disease or Multiple Sclerosis attending inpatient neurorehabilitation will be recruited to participate in this crossover study. Performance during a standardized gait adaptability (C-Gait) test on a VR-based treadmill (C-Mill+VR), as well as during clinical mobility tests (10-meter walk test, Timed Up & Go test, and stair ambulation) will be measured with and without an FFP2- mask (order randomized). In addition, participants will be asked about their perceived performance and perceived safety during the tests with and without a mask. Performance on the seven C-Gait subtests is based on centre of pressure-derived measures of foot placement in relation to the different tasks. These are averaged and added to a cognitive C-Gait task to give the overall composite score (primary outcome). Secondary outcomes will include the different subscores and clinical mobility tests.

This is a Registered Report and may have an associated publication; please check the article page on the journal site for any related articles.

**Data Availability Statement:** All relevant data from this study will be made available upon study completion.

**Funding:** The author(s) received no specific funding for this work.

**Competing interests:** The authors have declared that no competing interests exist.

## Potential significance

This study will make an important contribution to an ongoing debate regarding recommendations persons with and without a neurological disease should be given regarding wearing a face mask while walking. Furthermore, the study will complement the existing scientific discourse with clinical data from people with a neurological disease for whom falls, mobility deficits and mask wearing may be more frequent, which can help inform evidence-based recommendations.

## Trial registration

**German clinical trial register:** DRKS00030207.

## Background

During the COVID-19 pandemic, one of the most prevalent preventative measures is the wearing of face masks. This is recommended by the World Health Organization (WHO) and, for much of the pandemic, has been required in many indoor or crowded environments, especially when social distancing cannot be maintained. Any mask, whether cloth, surgical or special filtering masks (e.g. FFP2) protrudes into the lower visual field and, as a result, perception of visual stimuli in this area is reduced. The lower visual field provides important information for any necessary adjustments during walking (e.g., avoiding or stepping over obstacles) [1]. Therefore, the visual restriction caused by wearing a face mask could compromise safety and ultimately lead to a fall [2, 3].

A scientific debate regarding the recommendations that older adults should be given when wearing a face mask in order to minimize their fall risk took place recently after the Safe Exercise at Home website (endorsed by the Australian Physiotherapy Association) published recommendations for safe exercising in older adults in the COVID era. Using the motto "Mask Up, Look Down" [4] they advised older adults to look down more often while walking when wearing a mask, as doing so provides the visual information that is otherwise obtained through lower peripheral vision when looking ahead without wearing a mask. Kal et al. [5, 6] argued that such advice is flawed and can even be harmful as frequent and large amplitude movements of the head and the eyes could lead to a mismatch between visual and vestibular feedback and therefore cause instability and ultimately increase fall risk. Instead, Kal et al. recommended that older adults should try to walk more slowly [5, 6], as evidence suggests that this will give them more time to detect potential trip hazards and plan accordingly [7, 8]. Callisaya et al. [9] responded that tilting the head and directing the gaze towards the feet is a common strategy even for young adults [10] and raised concerns that that reducing walking speed may increase the risk of falls, citing studies that suggest stability of head and pelvis accelerations is optimised at one's comfortable walking speed [11] and that slower speeds may result in reduced gait quality [12]. In response, McCrum [13] highlighted the importance of the definition of stability. Citing additional studies [14–16], he argued that the mechanical stability of one's body configuration during walking in the anterior direction is increased when forward center of mass velocity is reduced meaning that, everything else being equal, reducing walking speed will directly reduce the likelihood of a forward fall following a trip.

As this recent debate highlighted, there is currently no clear consensus on the various factors that interact to influence walking safety while wearing a mask and all involved in the

described discussions agreed that systematic investigation of these issues is needed in order to move towards evidence-based recommendations. Since the risk of falling while wearing a mask is likely to be even greater in persons who already suffer from gait and balance impairments, especially those who are more reliant on vision [17], it is particularly urgent to address this issue in people with neurological diseases [18].

The first aim of this study is to investigate the effects of wearing a mask on objectively measured walking adaptability. Since perception and gait efficacy is known to have an influence on walking performance [19], the second aim of this study is to explore the relationships between wearing a face mask and gait efficacy and perceived safety. To address these aims, performance during a standardized test of walking adaptability (C-Gait) [20] on a VR-based treadmill (C-Mill+VR; Motek Medical, Amsterdam, The Netherlands) as well as during specific clinical tests (10-meter walk test (10MWT), Timed Up & Go test (TUG) and stair ambulation) will be measured with and without a face mask in 50 people with either Parkinson's disease or multiple sclerosis, two populations at a greatly increased risk of falls [21, 22]. In addition, participants will be asked about their perceived performance and perceived safety during the tests with and without a mask. We will test the hypothesis of a potentially meaningful decline (indicated by a moderate to large effect size; see "Sample Size Calculation" below) in the C-Gait composite score (primary outcome as an objective measure of walking adaptability) as a result of wearing a face mask. We will supplement this with exploratory analyses of the C-Gait subscores to gain insight into specific walking adaptability tasks and with exploratory analyses of the 10MWT, the TUG test and the stair negotiation test for which we also tentatively test the hypothesis of a medium to large effect.

## Methods

### Recruitment & experimental procedures

Participants will be persons that were prescribed inpatient rehabilitation and will be recruited during their stay at the Neurological Rehabilitation Center "Godeshoehe". We aim to include 50 participants. However, the medical director, as well as the executing researcher in communication with the ethics committee, might decide to terminate recruiting in certain circumstances (e.g., changes in the pandemic situation). Only participants that fulfil the clinic's standard aptitude checks for physical therapy and are not contraindicated (e.g. severe cognitive, visual or hearing impairment, > 135 kg bodyweight, > 2.00 meter body height, open skin lesion or bandage in the area of harness contact, ambulators with Functional Ambulation Categories (FAC) < 2, severe reduced bone density, spinal instability or unstable fractures, severe vascular disorders or cardiac abnormalities that affect the ability to exercise safely) with the use of the study's main assessment tool (C-Mill) upon admission will be asked if they would be interested to participate in the study. In case they are principally interested, they will be informed verbally and in writing about the study by the study coordinator (ME) and the neuropsychology staff. Participants will be given detailed written study information, which has been approved by the ethics committee. If they agree to participate, they will have to provide their written informed consent for participation and data processing. The study has been approved by the ethics committee of the medical faculty of the University of Bonn (File-Nr.: 269/21) and was registered in the German Clinical Trial Register (Registration number: DRKS00030207). All study procedures will be in compliance with the declaration of Helsinki.

Data collection is planned to start in 12/2022 and end in 7/2023.

In the morning of the study measurement day, participants will undergo the C-Gait Test. The C-Gait walking test will be completed once with and once without an FFP2 face mask by each participant at two different difficulty levels, as defined based on the C-Gait decision

algorithm by Timmermans et al. [20]. The order "mask/no mask" will be randomized per participant. Participants will be randomised to order 1 (wearing a face mask followed by not wearing a face mask) or order 2 (not wearing a face mask followed by wearing a face mask) by a paper lot that will be drawn from a sealed envelope by a person not related to the study. The lots will be prepared before participants' enrollment. Therefore, the numbers 1 to 50 will be randomised into the two study conditions with randomizer.org. Thereafter, paper-lots indicating a number and group allocation will be prepared and sealed in an envelope.

The C-Gait test is a standardised test that is integrated into the diagnostic program of the C-Mill by Motek VR treadmill (Motek Medical, Amsterdam, The Netherlands). It lasts 20 minutes and includes a series of seven sub-tasks to test walking adaptability.

Before starting the actual test, approximately 3 minutes of treadmill familiarization will take place, during which each participant's comfortable walking speed which will be determined according to Timmermans et al. [20] as follows: belt speed will be slowly increased (in steps of 0.1 km/h) until the participant reports it as comfortable. Subsequently, belt speed will be increased by 0.5 km/h followed by a stepwise decrease (0.1 km/h) until the participant reports it as comfortable again. These 2 indications of comfortable walking speed will then be averaged and taken to represent the participant's comfortable belt speed, which will be used for all sub-tasks. In the afternoon of the same day, participants will undergo the 10MWT, the TUG and the stair negotiation test. After each test, participants will be asked some questions regarding their perceived performance and perceived safety during the test with and without a face mask (see below). Testing and intervention won't be blinded, as the condition "mask/no mask") will be obvious to the assessors.

As modifications of the current protocol become necessary, they will be communicated within the research and the interdisciplinary clinical team as well as with the ethics committee (amendment) and the trial registration would be update accordingly.

## Inclusion & exclusion criteria

Patients will be included or excluded from the study according to the following criteria.

**Inclusion criteria.**

- diagnosed either by the treating physician or by a neurologist with either Parkinson's disease or multiple sclerosis

- willing to provide informed consent

- able to conduct the C-Gait Test

- inpatient of the Neurological Rehabilitation Center "Godeshoehe"

**Exclusion criteria.**

- MOCA score < 17

- other neurological or psychiatric illnesses that influence the ability to provide informed consent, understand the testing procedures, compromise safety during data collection or are known to influence motor functions

- insufficient knowledge of German hindering the ability to follow instructions and tests

- uncorrected visual or auditory impairments

## Study outcomes

The C-Gait Test measures walking adaptability using seven different subtests of the C-Mill + VR treadmill. During the test, participants are secured via a harness and their comfortable walking speed is determined at the start of the test. This speed is maintained throughout all subtests (by the system). In this study, the same walking speed will be used for the test with and without a mask.

Instructions are given both via an integrated screen and by the therapists. The seven different subtests include: 1) goal-directed stepping, 2) tandem walking, 3) obstacle avoidance, 4) slalom walking, 5) walking with suddenly shifting obstacles and targets, 6) speed adaptations, 7) dual-task walking (verbal Stroop task). A video showing examples of these tasks from Timmermans et al. (2019) can be found in the supplementary material of this article (S1 Video). All seven subtests of the C-Gait will be conducted first with an easy (level 2 of 5) and then with a difficult level (level 4 of 5) of difficulty. For each level (lowest: 1, highest: 5) the individual subtasks (except the cognitive task) are increased in difficulty as follows: 1) goal directed stepping: increased randomization of step length and width; 2) obstacle avoidance: increased size of obstacles and decreased available response time; 3) slalom walking: increased sharpness of the curves; 4) speed adaption: faster changes in speed; 5) tandem walking: decreased walking-area-width; and 6) walking with suddenly shifting obstacles and targets: increased size of obstacles and decreased available response time. According to standardized criteria, performance for each item is scored 0–100%. A detailed description of the C-Gait is provided elsewhere [20]. Performance of the five subtests "goal-directed stepping", "tandem walking", "obstacle avoidance", "slalom walking" and "speed adaptations" will be measured by the proportion of steps during which the center of pressure under the foot is within the projected area during the mid-stance phase (extended by half of the foot size for goal-directed stepping and obstacle avoidance). In the "walking with suddenly shifting obstacles and targets" task, the weighted mean of correct steps and correctly avoided obstacles is calculated. The cognitive dual-task is scored as the percentage of correct answers in a verbal Stroop task. The primary endpoint of the study is the overall performance of the C-Gait Test (composite score, calculated as Mean C-Gait scores on high level + C-Gait score for the cognitive performance task). This outcome parameter will be compared per condition (1: with face mask, 2: without face mask). Additionally, and also to account for participants who do not complete the high difficulty level, secondary endpoints including performance of the seven C-Gait subtests (calculated as Difficulty level x 2 x Performance (%) / 100), as well as performance (duration) of the four field tests (10 Meter Walk test, Timed Up-and-Go test, Timed Up-and Go test in dual task (serial 3s starting from 100) and Stair ascent/Stair Descent) will be assessed.

## Assessments

Sociodemographic and clinical data will be obtained by interview and self-report questionnaires. Motor testing will be performed by physiotherapists and/or exercise scientists with extended experience with the assessments in these patient groups. In case of any health issues (disease-related on unrelated) or any discomfort during testing, the assessments will be paused or terminated accordingly. Participants will be given the option to take breaks between tests whenever needed. All forms of adverse events during testing and intervention will be documented, reported (if necessary), communicated within the interdisciplinary team, and analyzed. Participants who experience culpably caused damage could assert a claim for compensation, which would be covered by the liability insurance of the rehabilitation center. Beside this there will be no additional insurance for participants. Insurance conditions are communicated to participants verbally and in the written study information.

The following assessments will be conducted:

- Expanded Disability Status Scale (EDSS) (score) [23]

- Activities-Specific Balance Confidence Scale (ABC) [24] (score)

- C-Gait Test [20] (composite score)

○ goal-directed stepping task (score)

○ tandem walking task (score)

○ obstacle avoidance task (score)

○ slalom walking task (score)

○ walking with suddenly shifting obstacles and targets task (score)

○ speed adaptations task (score)

○ dual-task walking (% of the correct answer at a verbal Stroop task)

- 10-meter Walk Test (10MWT) [25] (time, (s))

- Timed Up and Go Test (TUG) [26] (time, (s))

- Timed Up and Go Test in dual-tasking (TUG+DT) [27]

- Ascending/ descending stair test [28]

- Tailored questionnaire assessing perceived performance and perceived safety during the tasks with and without a mask

○ How did you perform on the gait tests with versus without a mask?

○ How safe did you feel during the gait tests with versus without a mask during the gait tests with or without wearing a face mask?

All study data will be pseudonymised during data collection and anonymized during the data entry processes. Only anonymized data will be published, presented, and made available to other researchers. All data collection, handling, and processing will be according current regulations (General Data Protection Regulation, German Federal Data Protection Act, Health Data Protection Act of North Rhine Westfphalia (NRW) and the revised Declaration of Helsinki (current version from 2013)). The planned procedures were checked by the clinic's data protection officer and were also submitted to the ethics committee. Participants are informed about data handling protocol in the written study description.

## Sample size

As this is an unfunded study in a clinical setting, certain resource constraints have been taken into consideration when planning the sample size. Considering the researcher and laboratory time available for the study and the number of patients available within its intended study duration, we estimate that fifty participants can feasibly be included in the study.

As the C-Gait, and VR treadmill-based gait adaptability assessments are a recent development, there is limited data available in the literature to make an accurate prediction of the expected size of the effect of face masks on these outcomes. Similarly, clinically meaningful differences or changes in the intended primary outcome parameters have not yet been established, due to the novelty of the topic. Two studies have compared groups with well-established balance and mobility differences from which we can draw some insight. Chen et al. [29] compared people with Parkinson's disease with and without freezing on the various subscales of the C-Gait and found

differences with effect sizes (Cohen's *d*) ranging from 0.43–0.92, with an average of 0.64. Timmermans et al. [20] compared the composite C-Gait scores between healthy adults (young and middle aged), healthy older adults, and older adults with balance and gait deficits. The effect sizes of the differences were 1.98 and 1.57 when comparing the healthy adults to the healthy older adults and older adults with balance and gait deficits, respectively, and a difference of *d* = 0.57 between the two older adult groups. Taken together, differences in C-Gait scores between groups with meaningful differences in mobility appear to be of medium to large magnitude. Therefore, if we were to observe medium or larger effect sizes for the difference in C-Gait scores between masked and unmasked assessments, we could infer that this change would be meaningful.

A sensitivity power analysis (S1 Fig) shows that with our feasible sample size, the analyses will have power of 0.8 to detect effect sizes of d = 0.4 and a power of 0.95 to detect effect sizes of d = 0.52, based on paired t-tests (within participants, mask vs. no mask). We believe that this, in combination with the various secondary parameters, will be sufficient to provide meaningful insight into the effect of face mask use on gait adaptability and mobility.

## Planned analyses

Data will be analysed using Jamovi [30] by EG and CM who will be blinded to the groups after recruitment of all participants. Interim analyses are not planned. Sample description will be done via frequencies and corresponding percentages, medians (with minimum and maximum), or means (with standard deviations). To verify suitability of parametric statistics, Shapiro-Wilk normality tests will be run for each outcome variable. If data are not normally distributed, the first and third quartiles as summaries of the dispersion, will be provided. To test the hypothesis of a potentially meaningful decline (indicated by a moderate to large effect size) in the C-Gait composite score (primary outcome as an objective measure of walking adaptability), paired t-tests (in case data is normally distributed) or Wilcoxon signed-rank tests (in case the data is not normally distributed) will be applied. To address the secondary exploratory hypotheses of an effect of face masks on specific C-Gait subtests and on the clinical tests, the same statistical approach will be applied. The test statistics, the p values and the effect size calculated as Cohen's *d* will be reported. Based on the limited literature described in our sample size justification above, we will treat effects larger than *d* = 0.57 as meaningful, though it is currently unknown if smaller effects would also be clinically important. For all, the significance level will be set at $p < 0.05$.

In case of missing data (e.g., due to measurement error or e.g., sudden illness or drop out) the respective participant will be excluded from that specific analysis, if complete data for others are available.

## Timeline

Data collection is planned to start in 12/2022 and end in 7/2023.

## Ethical approval and registration

The study was approved by the Ethics Committee of the University Bonn, Germany (269/21). The Study is registered in the German Register for Clinical Trials (Registration Number: DRKS00030207).

## Supporting information

**S1 Checklist. SPIRIT 2013 checklist: Recommended items to address in a clinical trial protocol and related documents*.**
(DOC)

**S1 Video. Examples of the C-Gait task from Timmermans et al. (2019) [20].**
(7Z)

**S1 Fig. Results of the sensitivity analysis.**
(TIF)

**S1 File.**
(DOCX)

**S2 File.**
(DOCX)

## Acknowledgments

We thank Professor Hans Karbe, Jochen Saliger, and Florian Wolf of the Neurological Rehabilitation Center Godeshoehe for their support in preparing this research project.

## Author Contributions

**Conceptualization:** Mareike Eschweiler, Christopher McCrum, Eleftheria Giannouli.

**Methodology:** Mareike Eschweiler, Christopher McCrum, Eleftheria Giannouli.

**Project administration:** Mareike Eschweiler.

**Resources:** Mareike Eschweiler.

**Supervision:** Christopher McCrum, Eleftheria Giannouli.

**Writing – original draft:** Mareike Eschweiler.

**Writing – review & editing:** Christopher McCrum, Eleftheria Giannouli.

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
