## [Decision Letter · Decision Letter 0]

27 Jan 2023

PONE-D-22-24759A registered report of a crossover study on the effects of face masks on walking adaptability in people with Parkinson’s disease and multiple sclerosisPLOS ONE

Dear Dr. Giannouli,

Thank you for submitting your manuscript to PLOS ONE. After careful consideration, we feel that it has merit but does not fully meet PLOS ONE’s publication criteria as it currently stands. Therefore, we invite you to submit a revised version of the manuscript that addresses the points raised during the review process.

Overall this is an interesting and relevant manuscript.  The reviewers have all raised some important points after reviewing this manuscript, but those should be easily addressed.  Based on the existing literature about vision and walking considering the potential impact on the look ahead window the point about measuring head pitch during the walking task is important to address.

We look forward to receiving your revised manuscript.

Kind regards,

Eric R. Anson

Academic Editor

PLOS ONE

and https://journals.plos.org/plosone/s/file?id=ba62/PLOSOne_formatting_sample_title_authors_affiliations.pdf.

2. In your cover letter, please confirm that the research you have described in your manuscript, including participant recruitment, data collection, modification, or processing, has not started and will not start until after your paper has been accepted to the journal (assuming data need to be collected or participants recruited specifically for your study). In order to proceed with your submission, you must provide confirmation.

Reviewers' comments:

Reviewer's Responses to Questions

**Comments to the Author**

1. Does the manuscript provide a valid rationale for the proposed study, with clearly identified and justified research questions?

Reviewer #1: Yes

Reviewer #2: Yes

Reviewer #3: Partly

2. Is the protocol technically sound and planned in a manner that will lead to a meaningful outcome and allow testing the stated hypotheses?

Reviewer #1: Yes

Reviewer #2: Partly

Reviewer #3: No

3. Is the methodology feasible and described in sufficient detail to allow the work to be replicable?

Reviewer #1: Yes

Reviewer #2: Yes

Reviewer #3: No

4. Have the authors described where all data underlying the findings will be made available when the study is complete?

Reviewer #1: Yes

Reviewer #2: Yes

Reviewer #3: Yes

5. Is the manuscript presented in an intelligible fashion and written in standard English?

Reviewer #1: Yes

Reviewer #2: Yes

Reviewer #3: No

6. Review Comments to the Author

You may also provide optional suggestions and comments to authors that they might find helpful in planning their study.

Reviewer #1: In this registered protocol report, the aim is to investigate the effects of mask-wearing on walking adaptability in people with Parkinson’s disease and Multiple Sclerosis. Gait measures with and without a mask will be collected and will be compared using t-tests or Wilcoxon signed-rank tests.

Minor revisions:

1- The standard statistical term for average is mean.

2- Line 300: In addition to frequencies, provide the corresponding percentage.

3- Line 300: Typically the first and third quartiles are provided as summaries of the dispersion when data is not not normally distributed.

Reviewer #2: This manuscript is about a registered report of a study that will investigate the influence of wearing a facial mask on several gait tests in Parkinson’s and Multiple Sclerosis patients. This is an interesting investigation because the facial mask causes a loss of part of the lower visual field, which can compromise gait adaptability and increase the risk of falls. This study is an initial response to the scientific debate raised regarding the recommendations that older adults should be given when wearing a face mask.

Although this study is well designed, there is no measure related to head pitch in the walking tasks. It would be relevant to add this parameter, especially for the c-mill walking tasks, as the participants may compensate for the loss of the lower visual field by pitching down the head to acquire the necessary visual information to perform the tasks successfully.

Also, no measure captures the amount of lower visual field loss due to mask-wearing. Some measures related to that would also help understand how the mask may affect gait performance.

Regarding the inclusion criteria, would the ability to walk independently without an aid device be a requirement to participate in this study? In addition, would Parkinson’s disease patients have a Hoehn and Yahr between 1 and 3? How about the Multiple Sclerosis patients – would you use a scale to define participants’ disability status?

Minor comments

- line 179: what does FAC mean?

- line 180: How will “severe reduced bone density” be assessed?

Reviewer #3: Dear authors,

I read your paper with interest and here I am writing some points for you.

Please doublecheck the writing of your paper.

The tense of presentation is different in sections.

You are talking about increasing the rate of falling because of wearing mask among patients with Parkinson or MS. Is it evidence based? Please let us know about published evidences in this regard.

Your test was done on a isolated walking area, treadmill. How you can compare it with walking in street or at home?? How you have considered the environmental parameters?

In potential significant section, you suddenly focus on older adults.

" fall risk took place recently after the Safe Exercise at Home website". I am not sure about the mechanism happened here. Maybe in the first session they have some difficulties, not in all the time. Home based exercises are simple and easy to do and there is not much mobility.

I got confused, because in some parts you talk about older adults and in some parts you talk about patients with Parkinson/MS. Please be careful. A patient with MS could be 25 years old, so you can not consider her as an older adult.

Then you give some evidence regarding younger adults.

There is not a consistency in your presentation in Introduction. In addition the novelty of work is not highlighted. It is not exactly in line with your main aim and target group.

Method section should be rewritten based on an standard format.

For example: However, the medical director, as well as the executing researcher in communication with the ethics committee, might decide to terminate recruiting in certain circumstances(e.g., changes in the pandemic situation).

Recruitment & Experimental Procedures is too long and messy.

In inclusion criteria, the what stage of MS is acceptable for you? This section needs to more accurate details. What about their medicines?

I think it is a protocol of study. Is it? Please double check your title.

Your figures are not clear. Are those necessary?

Once you answer to those general points, then I will be able review your paper accurately.

Good luck

7. PLOS authors have the option to publish the peer review history of their article (what does this mean?). If published, this will include your full peer review and any attached files.

Reviewer #1: No

Reviewer #2: No

Reviewer #3: No

---

## [Author Response · Author response to Decision Letter 0]

13 Mar 2023

Replies to the academic editor’s and reviewers’ comments

The authors would like to thank the editor and the reviewers for taking the time to read our work and for the helpful comments which will improve the manuscript. Some good points have been brought up and we appreciate the opportunity to clarify our research methodology. As you will see, we revised our manuscript taking into consideration the comments of the reviewers and the editor. Below you will find the concrete answers to each comment. 

We will use abbreviations for persons with Parkinson’s disease (pwPD) and persons with multiple sclerosis (pwMS).

Below we include the editor and reviewer comments in italics, followed by our responses.

Academic editor

1. Based on the existing literature about vision and walking considering the potential impact on the look ahead window the point about measuring head pitch during the walking task is important to address. 

• We thank the editor for highlighting this issue and agree that it is one of the most important points raised by the reviewers. Please see our response on this point to reviewer 2 below (comment 5). Note that we are specifically requesting the editor and reviewers’ opinions on the possibilities here, since the registered report format allows us this input before conducting the study.

Reviewer #1

2. The standard statistical term for average is mean

• Thank you for bringing this to our attention. We exchanged terms in L210.

3. Line 300: In addition to frequencies, provide the corresponding percentage.

• We will be happy to provide the corresponding percentage, as you suggested, since we agree this will enhance the value of information. We made changes accordingly see L293.

4. Line 300: Typically, the first and third quartiles are provided as summaries of the dispersion when data is not normally distributed. 

• You are right, reporting the first and third quartiles as summaries of the dispersion when data is not normally distributed, is the standard procedure. Therefore, we will provide these. We added a sentence on that in the manuscript L295-295.

Reviewer #2

5. Although this study is well designed, there is no measure related to head pitch in the walking tasks. It would be relevant to add this parameter, especially for the c-mill walking tasks, as the participants may compensate for the loss of the lower visual field by pitching down the head to acquire the necessary visual information to perform the tasks successfully. 

• Thank you very much for the compliments on our design. We agree that the possibility of pitching the head might be an expectable form of compensation that we should consider. In addition to the suggestion that this may allow participants to maintain their performance across conditions, we would also like to point out that increased forward head pitch may also have a negative effect on gait and balance due to the more anterior center of mass position that this will result in and could, therefore, also lead to a decline in performance. As this is an applied study in a clinical setting, we only have the resources and aim to investigate the overall outcome, not the specific mechanisms, since more detailed 3D motion capture will not be possible. In the clinical setting, we would suggest that the practically relevant result is initially whether or not the condition results in a reduction in performance. Should we find or not find a difference, future research can more precisely look into these changes with more appropriate, controlled setups. This being said, there would be a few options to gain some insight into this issue and given the registered report format, we would like to offer these ideas to the editor and reviewers before adjusting our protocol. 

o Option 1: Add a 5-point visual Likert scale questionnaire asking participants to rate their head pitch while completing the task with the mask compared to without the mask (“I felt like I tilted my head forward more while wearing the mask: 1 Strongly disagree, 2 disagree, 3 neutral/not sure, 4 agree, 5 strongly agree.) This will not objectively measure head tilt, but will provide information on whether or not participants perceived a change in their head position to complete the task and this may indicate that changes did or did not take place. Moreover, add a 5-point Likert scale questionnaire asking the assessing researcher to rate the patients’ head tilt during each of the tasks.

o Option 2: Take video camera recordings of the trials from a standardized position relative to the C-Mill treadmill. Have two or three of the researchers rate the head tilt during each trial as described above for a more robust estimation and to check reliability of the procedure.

o Option 3: Take video recordings as described above and use 2D video analysis to label and track head tilt for a more objective outcome. 

We would like to point out that while option 3 seems to be the most objective approach, none of the authors have expertise or experience in this type of analysis and it the accuracy will not be as high as it would be with 3D motion capture. Given the significant amount of time that will be needed to label and analyze the videos for this approach and the potentially limited accuracy, we are uncertain if this will really be a worthwhile approach. We would therefore be pleased to hear the opinions of the reviewers and editors on these approaches and what they would find acceptable for the study given the clinical setting and context.

6. Also, no measure captures the amount of lower visual field loss due to mask-wearing. Some measures related to that would also help understand how the mask may affect gait performance. 

• Thank you for your advice on this aspect. Since this a first experimental study on that topic with limited resources, we are unfortunately not able to take all measures into account. Existing evidence suggests 9.5-13% of negative effect on visual-field test performance (1, 2), which could be reduced to 6% when fitting the superior border of the mask well to the person’s face (1). Therefore, in our study we will try to ensure a good fit of the mask. If we still find a deficit in performance while wearing the mask, it is likely that a small percentage of visual field loss was present. We will take this evidence into account, when interpreting our results.

7. Regarding the inclusion criteria, would the ability to walk independently without an aid device be a requirement to participate in this study? In addition, would Parkinson’s disease patients have a Hoehn and Yahr between 1 and 3? How about the Multiple Sclerosis patients – would you use a scale to define participants’ disability status? 

• We would like to thank this reviewer for giving us the chance to clarify this. All pwPD and pwMS that are admitted in the clinic, in which the study will take place, undergo the following tests as part of the clinic`s standard procedures: standing with narrow stance and open eyes, standing with narrow stance and closed eyes and walking for 1 min. Patients who are able to perform these tests independently or with standby assistance, will be considered for our study. Based on existing clinical data, such persons are on average: pwPD with a Hoehn & Yahr stage 1-3 and pwMS with an EDSS of 0.0-4.0. We will record and report these stages of the participants in the descriptive results. 

8. line 179: what does FAC mean? 

• Sorry for not introducing this abbreviation. FAC means Functional Ambulation Categories (3-5) and refers to a scale rating peoples walking ability by five categories, where low scores indicate less ability to walk and higher scores more independent ambulation. (0: Non-functional ambulatory, 1: Ambulator, dependent on continuous physical assistance, 2: Ambulator, dependent on intermittent physical assistance, 3: Ambulator, dependent on supervision, 4: Ambulator, independent level surface only, 5: Ambulator, independent). We introduced the abbreviation in L127-128.

9. line 180: How will “severe reduced bone density” be assessed?

• Apologies for the confusion. Severe reduced bone density, spinal instability or unstable fractures, and severe vascular disorders or cardiac abnormalities that affect the ability to exercise safely are general contraindications according to the C-Mill manufacture manual. These will not be explicitly assessed in our study, but patients’ medical and anamnestic reports will serve as a source for this information. Patients with those contraindicators will not be eligible for participation in our study. We clarified this in L125-130. 

Reviewer #3

10. Please doublecheck the writing of your paper. The tense of presentation is different in sections. 

• This manuscript is a registered report protocol for a study that is not yet conducted (stage one being very similar to a typical protocol article). 

11. You are talking about increasing the rate of falling because of wearing mask among patients with Parkinson or MS. Is it evidence based? Please let us know about published evidences in this regard. 

• We would like to clarify, that we do not state that pwPD and pwMS fall more often because they wear a face mask. We provide evidence that occlusion of the lower visual field might increase the risk to stumble and eventually fall (6,7). We also highlighted that there is no clear consensus on the various factors that interact to influence walking safety while wearing a mask (8-20). Based on the discussed evidence, those persons who already suffer from impairments in balance and gait, being more dependent visual information (21) and already having an increased fall risk like pwPD (22) and pwMS (23) might have an even enhanced risk for falls when wearing a mask. We argue that it is therefore important to investigate this clinically in a neurological cohort, which is supported by (24). It is the aim of our study to further investigate this question within an inpatient clinical setting.

12. Your test was done on a isolated walking area, treadmill. How you can compare it with walking in street or at home?? How you have considered the environmental parameters?

• Our primary assessment will be done on a treadmill with virtual reality so we are able to reliably simulate environmental situations and obtain objective measurements of walking adaptability based on center of pressure data. As secondary outcomes, we will also assess typical clinical mobility parameters with and without wearing a face mask, that are commonly applied to draw conclusions on real life mobility (10 metre walk test, timed-up-and-go test, stair ascent and descent). Additionally, patients will be asked to reflect on their performance and safety with and without wearing a face mask. The subjective performance and safety perception might also be indicative for subjective fears/ carelessness related to patients’ mobility. We believe that this combination of tasks and outcomes is appropriate to address our aims.

13. In potential significant section, you suddenly focus on older adults.

• The ongoing debate only focused on older adults and we will add knowledge on persons with a neurological disease. We replaced “older adults” by “persons with and without a neurological disease” to enhance clarity, see L46. 

14. " fall risk took place recently after the Safe Exercise at Home website". I am not sure about the mechanism happened here. Maybe in the first session they have some difficulties, not in all the time. Home based exercises are simple and easy to do and there is not much mobility.

• The citation you refer to in your comment needs to be considered as a whole: “A scientific debate regarding the recommendations that older adults should be given when wearing a face mask in order to minimize their fall risk took place recently after the Safe Exercise at Home website (endorsed by the Australian Physiotherapy Association) published recommendations for safe exercising in older adults in the COVID era.” We bring up this publication as it was the starting point for the scientific debate on what advice to give to older people for safe walking while wearing a face mask. This controversial debate shows that there is no clear consensus on the various factors that interact to influence walking safety while wearing a mask. Which for us was one of the reasons to look into that topic for persons who might be even more impaired in balance and gait, a neurological cohort.

15. I got confused, because in some parts you talk about older adults and in some parts you talk about patients with Parkinson/MS. Please be careful. A patient with MS could be 25 years old, so you cannot consider her as an older adult.

Then you give some evidence regarding younger adults.

There is not a consistency in your presentation in Introduction. In addition the novelty of work is not highlighted. It is not exactly in line with your main aim and target group.

• see our response to comment 11

16. Method section should be rewritten based on an standard format.

For example: However, the medical director, as well as the executing researcher in communication with the ethics committee, might decide to terminate recruiting in certain circumstances(e.g., changes in the pandemic situation).

• Since this is a Stage 1 Registered Report manuscript, we have written and structured the manuscript based on both the PLOS One guidelines for this article format and other general guidelines for registered reports. Please note that these typically include much more a priori methodological and study design details in the interest of transparency and accurate preregistration of the approach to be taken.

17. Recruitment & Experimental Procedures is too long and messy.

• See our response to the previous comment.

18. In inclusion criteria, the what stage of MS is acceptable for you? This section needs to more accurate details. What about their medicines?

• See our response to comment 7. 

19. I think it is a protocol of study. Is it? Please double check your title.

• This is a stage 1 registered report manuscript, which is indeed similar to a traditional protocol article. Since it is the stage 2 version that is eventually published, which includes results, discussion and conclusions, we will retain the current title to accommodate for the stage 2 version. 

20. Your figures are not clear. Are those necessary?

• Thank you for your comment and question on our figures. These are the results of the sensitivity analysis. PLOS One author guidelines request sample size estimations in registered report. Considering your question, we will provide the figure in the supplement. Changes to the manuscript were made accordingly, see L284 and L436.

References

1. Boxrud, C. A., Householder, N. A., Kim, D. K., Kugler, K. M., Harris, C. S., Benjamin, B. P., Panrudkevich, A. H., & Bahadur, G. G. (2023). Inferior altitudinal visual loss and mask-wearing practices: A case series. Indian journal of ophthalmology, 71(2), 657–660. https://doi.org/10.4103/ijo.IJO_934_22

2. Gómez Mariscal, M., Muñoz-Negrete, F. J., Muñoz-Ramón, P. V., Aguado Casanova, V., Jaumandreu, L., & Rebolleda, G. (2022). Avoiding mask-related artefacts in visual field tests during the COVID-19 pandemic. The British journal of ophthalmology, 106(7), 947–951. https://doi.org/10.1136/bjophthalmol-2020-318408

3. Holden, M. K., Gill, K. M., & Magliozzi, M. R. (1986). Gait assessment for neurologically impaired patients. Standards for outcome assessment. Physical therapy, 66(10), 1530–1539. https://doi.org/10.1093/ptj/66.10.1530

4. Holden, M. K., Gill, K. M., Magliozzi, M. R., Nathan, J., & Piehl-Baker, L. (1984). Clinical gait assessment in the neurologically impaired. Reliability and meaningfulness. Physical therapy, 64(1), 35–40. https://doi.org/10.1093/ptj/64.1.35

5. Marks, D. (2011). Gehfähigkeit: Functional Ambulation Categories (FAC). In Schädler, S., Kool, J., Lüthi, H., Marks, D., Oesch, P., Pfeffer, A., Wirz, M. Assessments in der Neurorehabilitation. Band 1: Neurologie, 3rd Ed. Bern, Hans Huber Verlag

6. Buckley JG, Timmis MA, Scally AJ, Elliott DB. When Is Visual Information Used to Control Locomotion When Descending a Kerb? PLoS One [Internet]. 2011 [cited 2021 Sep 10];6(4):e19079. Available from: https://journals.plos.org/plosone/article?id=10.1371/journal.pone.0019079

7. Rietdyk S, Rhea CK. The effect of the visual characteristics of obstacles on risk of tripping and gait parameters during locomotion. Ophthalmic Physiol Opt [Internet]. 2011 May 1 [cited 2021 Sep 10];31(3):302–10. Available fr

8. Association AP. Mask up and look down | Safe Exercise at Home [Internet]. 2021 [cited 2021 Sep 12]. Available from: https://www.safeexerciseathome.org.au/mask-up-and-look-down

9. Kal EC, Young WR, Ellmers TJ. Face masks, vision, and risk of falls. BMJ [Internet]. 2020 Oct 28 [cited 2020 Nov 2];371:m4133. Available from: https://www.bmj.com/lookup/doi/10.1136/bmj.m4133

10. Kal EC, Young WR, Ellmers TJ. Authors’ response to: “Face masks and risk of falls – a vision for personalised advice and timing?” by Callisaya et al. BMJ. 2020; 

11. Curzon-Jones BT, Hollands MA. Route previewing results in altered gaze behaviour, increased self-confidence and improved stepping safety in both young and older adults during adaptive locomotion. Exp Brain Res 2018 2364 [Internet]. 2018 Feb 13 [cited 2021 Sep 12];236(4):1077–89. Available from: https://link.springer.com/article/10.1007/s00221-018-5203-9

12. Ellmers TJ, Cocks AJ, Young WR. Evidence of a Link Between Fall-Related Anxiety and High-Risk Patterns of Visual Search in Older Adults During Adaptive Locomotion. Journals Gerontol Ser A [Internet]. 2020 Apr 17 [cited 2021 Sep 12];75(5):961–7. Available from: https://academic.oup.com/biomedgerontology/article/75/5/961/5541624

13. Callisaya ML, Hill K, Hill A-M, Mackintosh S, Said CM, Sherrington C, et al. Face masks and risk of falls – a vision for personalised advice and timing? BMJ. 2020;(4):8–10. 

14. Marigold DS, Patla AE. Visual information from the lower visual field is important for walking across multi-surface terrain. Exp Brain Res 2008 1881 [Internet]. 2008 Mar 6 [cited 2021 Sep 12];188(1):23–31. Available from: https://link.springer.com/article/10.1007/s00221-008-1335-7

15. Latt MD, Hylton AE, Menz B, Victor AE, Fung S, Lord SR. Walking speed, cadence and step length are selected to optimize the stability of head and pelvis accelerations. 

16. Huijben B, van Schooten KS, van Dieën JH, Pijnappels M. The effect of walking speed on quality of gait in older adults. Gait Posture. 2018 Sep 1;65:112–6. 

17. McCrum C. Walking slower increases anterior stability to a trip: a consideration for face masks and falls risk. BMJ. 2020;1–2. 

18. Pavol MJ, Owings TM, Foley KT, Grabiner MD. Gait Characteristics as Risk Factors for Falling From Trips Induced in Older Adults. Journals Gerontol Ser A [Internet]. 1999 Nov 1 [cited 2021 Sep 12];54(11):M583–90. Available from: https://academic.oup.com/biomedgerontology/article/54/11/M583/544811

19. Süptitz F, Karamanidis K, Catalá MM, Brüggemann GP. Symmetry and reproducibility of the components of dynamic stability in young adults at different walking velocities on the treadmill. J Electromyogr Kinesiol. 2012 Apr 1;22(2):301–7. 

20. McCrum C, Willems P, Karamanidis K, Meijer K. Stability-normalised walking speed: A new approach for human gait perturbation research. J Biomech. 2019 Apr 18;87:48–53.

21. Yakubovich S, Israeli-Korn S, Halperin O, Yahalom G, Hassin-Baer S, Zaidel A. Visual self-motion cues are impaired yet overweighted during visual–vestibular integration in Parkinson’s disease. Brain Commun [Internet]. 2020 Jan 1 [cited 2021 Sep 10];2(1). Available from: https://academic.oup.com/braincomms/article/2/1/fcaa035/5814201

22. Nilsagard Y, Gunn H, Freeman J, Hoang P, Lord S, Mazumder R, et al. Falls in people with MS - An individual data meta-analysis from studies from Australia, Sweden, United Kingdom and the United States. Mult Scler J [Internet]. 2015 Jan 14 [cited 2021 Nov 30];21(1):92–100. Available from: https://journals.sagepub.com/doi/10.1177/1352458514538884

23. Homann B, Plaschg A, Grundner M, Haubenhofer A, Griedl T, Ivanic G, et al. The impact of neurological disorders on the risk for falls in the community dwelling elderly: a case-controlled study. BMJ Open [Internet]. 2013 Nov 1 [cited 2021 Nov 30];3(11):e003367. Available from: https://bmjopen.bmj.com/content/3/11/e003367

24. Klatt BN, Anson ER. Navigating Through a COVID-19 World: Avoiding Obstacles. J Neurol Phys Ther [Internet]. 2021 Jan 1 [cited 2021 Mar 30];45(1):36–40. Available from: https://journals.lww.com/10.1097/NPT.0000000000000338

---

## [Decision Letter · Decision Letter 1]

16 May 2023

A registered report of a crossover study on the effects of face masks on walking adaptability in people with Parkinson’s disease and multiple sclerosis

PONE-D-22-24759R1

Dear Dr. Giannouli,

We’re pleased to inform you that your manuscript has been judged scientifically suitable for publication and will be formally accepted for publication once it meets all outstanding technical requirements.

Kind regards,

Eric R. Anson

Academic Editor

PLOS ONE

Additional Editor Comments (optional):

Reviewers' comments:

Reviewer's Responses to Questions

**Comments to the Author**

1. Does the manuscript provide a valid rationale for the proposed study, with clearly identified and justified research questions?

Reviewer #1: Yes

Reviewer #2: Yes

Reviewer #3: Yes

2. Is the protocol technically sound and planned in a manner that will lead to a meaningful outcome and allow testing the stated hypotheses?

Reviewer #1: Yes

Reviewer #2: Yes

Reviewer #3: Yes

3. Is the methodology feasible and described in sufficient detail to allow the work to be replicable?

Reviewer #1: Yes

Reviewer #2: Yes

Reviewer #3: Yes

4. Have the authors described where all data underlying the findings will be made available when the study is complete?

Reviewer #1: Yes

Reviewer #2: Yes

Reviewer #3: Yes

5. Is the manuscript presented in an intelligible fashion and written in standard English?

Reviewer #1: Yes

Reviewer #2: Yes

Reviewer #3: Yes

6. Review Comments to the Author

You may also provide optional suggestions and comments to authors that they might find helpful in planning their study.

Reviewer #1: All comments relating to the statistical aspects of the manuscript have been adequately addressed.

Reviewer #2: I want to thank the authors for thoughtfully revising the manuscript. I understand the limitations of clinical studies like this regarding using a 3D motion analysis system to assess head pitch angle. I liked that the authors proposed three different options to introduce simple measures of head motion. I agree with the authors that option 3 is the most objective, whereas, in my opinion, option 1 is the most subjective. I would go with options 2 or 3. For option 3, you do not need to analyze the entire time series. You could measure head tilt at foot contact. This should be sufficient to get a reasonable estimate of changes due to mask-wearing. As a suggestion, you could try the software Kinovea (https://www.kinovea.org/).

Reviewer #3: Dear authors,

Many thanks for your valuable submission.

You have done my comments accurately and I am happy about this revised version.

Kind regards

7. PLOS authors have the option to publish the peer review history of their article (what does this mean?). If published, this will include your full peer review and any attached files.

Reviewer #1: No

Reviewer #2: No

Reviewer #3: No

---

## [Editor Report · Acceptance letter]

16 Jun 2023

PONE-D-22-24759R1 

A registered report of a crossover study on the effects of face masks on walking adaptability in people with Parkinson’s disease and multiple sclerosis 

Dear Dr. Giannouli:

I'm pleased to inform you that your manuscript has been deemed suitable for publication in PLOS ONE. Congratulations! Your manuscript is now with our production department. 

Kind regards, 

on behalf of

Dr. Eric R. Anson 

Academic Editor

PLOS ONE